# A Yeast Mitotic Tale for the Nucleus and the Vacuoles to Embrace

**DOI:** 10.3390/ijms24129829

**Published:** 2023-06-06

**Authors:** Silvia Santana-Sosa, Emiliano Matos-Perdomo, Jessel Ayra-Plasencia, Félix Machín

**Affiliations:** 1Research Unit, University Hospital Ntra Sra de Candelaria, Ctra del Rosario 145, 38010 Santa Cruz de Tenerife, Spain; silvia.sansos@gmail.com (S.S.-S.); ematospe@ull.edu.es (E.M.-P.); jessel.ayra.plasencia@gmail.com (J.A.-P.); 2Institute of Biomedical Technologies, University of La Laguna, 38200 San Cristóbal de La Laguna, Spain; 3Faculty of Health Sciences, Fernando Pessoa Canarias University, 35450 Santa María de Guía, Spain

**Keywords:** mitosis, ribosomal DNA (rDNA), nucleolus, nuclear envelope, vacuole, lysosome, TOR, lipid metabolism, autophagy, nucleophagy

## Abstract

The morphology of the nucleus is roughly spherical in most eukaryotic cells. However, this organelle shape needs to change as the cell travels through narrow intercellular spaces during cell migration and during cell division in organisms that undergo closed mitosis, i.e., without dismantling the nuclear envelope, such as yeast. In addition, the nuclear morphology is often modified under stress and in pathological conditions, being a hallmark of cancer and senescent cells. Thus, understanding nuclear morphological dynamics is of uttermost importance, as pathways and proteins involved in nuclear shaping can be targeted in anticancer, antiaging, and antifungal therapies. Here, we review how and why the nuclear shape changes during mitotic blocks in yeast, introducing novel data that associate these changes with both the nucleolus and the vacuole. Altogether, these findings suggest a close relationship between the nucleolar domain of the nucleus and the autophagic organelle, which we also discuss here. Encouragingly, recent evidence in tumor cell lines has linked aberrant nuclear morphology to defects in lysosomal function.

## 1. The Yeast Nucleus: A Model to Address Nuclear Morphology and Architecture

### 1.1. Constituents of the Nuclear Envelope with a Role in Shaping the Nucleus

The main feature that characterizes eukaryotes is the presence of the nucleus, a membranous organelle that protects the genetic material from the rest of the processes that take place in the cytoplasm. The nucleoplasm is separated from the cytoplasm by a double membrane known as the nuclear envelope (NE). The outer layer of the NE (the outer nuclear membrane, ONM) is a continuum with the endoplasmic reticulum (ER) and it is also referred to as the perinuclear ER, although the ONM possesses unique proteins that distinguish it from the ER [1].

In most eukaryotes, there is a network of proteins under the inner nuclear membrane (INM) that constitute a mesh called the nuclear lamina, which provides structural support to the nucleus and contributes to chromatin organization, DNA repair, and transcriptional regulation. This nuclear lamina is composed of different types of proteins, including a variety of filamentous lamins as its main component as well as lamin-associated proteins, some of which are embedded in the INM [2,3]. However, unicellular organisms and plants do not have lamins, although they have other INM proteins that resemble lamins, either structurally or functionally [1]. In *S. cerevisiae*, one of the most studied unicellular eukaryotes, and the focus of this review, there are neither lamina nor lamins, yet some proteins may also functionally resemble lamins. One such protein is the coiled-coil Esc1, which is involved in anchoring silenced telomeres to the NE, and whose overexpression triggers NE expansion [4,5,6]. Although Esc1 is only present in yeast, Ebp2 and Rrs1, which are also lamin-related proteins that play key roles in telomere clustering and ribosome biogenesis in budding yeast, are universally conserved and have been related to several human diseases [7].

Communication between the nucleoplasm and the cytoplasm occurs through nuclear pores in the NE, in which the nuclear pore complexes (NPCs) are assembled to control molecular trafficking. In budding yeast, NPCs are massive assemblies of approximately 550 nucleoporins of ~30 different types that form a toroidal structure symmetrical around a central channel with flexible filaments on either side that serve as chromatin and cargo binding sites [8,9,10,11]. The number of NPCs varies throughout the cell cycle together with the nuclear surface, ranging from ~65 in G1 to ~200 in mitotic cells in *S. cerevisiae* haploid strains [12]. Their general structure and part of their components remain conserved from yeast to humans, although recent evidence suggests that some elements vary substantially between species [11,13,14]. Far from being mere support structures, they have been shown to be involved in processes such as alternative DNA recombination/repair, recombination-mediated telomere elongation, and gene expression regulation, while also acting as regulatory hubs for the coordination of transcript elongation, processing, and export [15,16,17].

Spindle pole bodies (SPBs) constitute the yeast analogue to centrosomes and, in contrast to the latter, they are embedded in the NE, from which the mitotic spindle arises while astral microtubules nucleate outward into the cytoplasm [18,19]. During interphase, the centromeres of yeast chromosomes are attached to the SPB by short interphase microtubules, forming the centromere cluster, which disassembles upon microtubule depolymerization [20,21,22,23]. The single SPB found in G1 cells is located opposite the nucleolus and is aligned with the site of new bud emergence [23,24,25,26]. The yeast SPB duplicates in S phase and partitions during mitosis, migrating to NE opposite poles in the mother–daughter axis [27].

The linker of the nucleoskeleton and cytoskeleton (LINC) complex joins nuclear and cytosolic elements and is highly conserved from yeast to mammals. It is composed of SUN (Sad1, UNC-84) and KASH (Klarsicht, Anc-1, Syne-1 homology) domain-containing proteins, with the former facing the nucleoplasm and the latter facing the cytoplasm. In yeast, only one SUN domain protein has been identified, Mps3, which forms a chromatin tether to the INM in addition to a structural role in the SPB [28,29]. It is believed to pair with Csm4 or Mps2, which are KASH-like proteins, forming the cytoskeletal link [30]. The cohibin complex, composed of Csm1 and Lrs4, mediates the interaction between Mps3 and the telomeres [31,32]. On the other hand, there is another group of INM proteins that interact with lamins or lamin-like proteins through a helix-extension-helix motif containing the LEM domain (LAP2, Emerin, MAN1), providing structural stability to the nucleus, chromosome attachment, and silencing regulation [33]. In the budding yeast, the MAN1 homologs Heh1 (also known as Src1) and Heh2 regulate NPC quality and stability [34,35]. Furthermore, Heh1 belongs to the CLIP (chromosome linkage INM proteins) complex that, together with the cohibin and RENT (regulator of nucleolar silencing and telophase exit) complexes, attaches the ribosomal DNA (rDNA) to the nucleolar periphery [31,36].

The role of the SPB in shaping the nucleus becomes evident during the nuclear division in this closed mitosis organism. The NE undergoes drastic morphological changes at this stage, which are essential to ensure a proper partition of the genetic material and the rest of the nucleoplasmic content between the progeny. The SPBs, as centers of microtubule attachment, ensure the enlargement of the nucleus driven by the spindle elongation and the pulling forces of astral microtubules [37]. Interestingly, the LINC and the SPB spatially co-localize in mitotic cells, whereas the LINC only spreads over the NE in meiosis, where it forms a specialized complex that connects telomeres to the cytoskeleton [38]. Mutants for both the LINC and SPBs share phenotypes of misshapen nuclei [39]. In addition, NPC mutants also lead to impressive aberrant NE morphologies as well [40]. Likewise, the other aforementioned INM proteins involved in chromatin tethering and organization play roles in the maintenance of a proper nuclear shape [41]. Homologs of these proteins in higher eukaryotes also participate in maintaining the right shape of the nucleus; however, it is the lamina that has attracted more attention because a mutation in the LMNA (lamin A) gene causes the most outstanding premature aging disease in humans, the Hutchinson–Gilford Progeria Syndrome [42]. 

### 1.2. Physiological Changes in Nuclear Shape and Structure

Most nuclei in eukaryotes are rounded/oval shaped, and loss of sphericity has been associated with cancer and aging [43,44]. In contrast, an altered nuclear shape is essential for certain cell types to exert their function, such as cells of the immune system. For instance, neutrophils possess multilobulated nuclei (from three to four lobes in humans), and this organization correlates with their ability to migrate through the vascular endothelium and reach the potential hazard. Changes in the pattern of lamin expression, together with a different NE lipid composition, are key to attaining this lobulated shape [45,46,47]. The physiological impact of proper nuclear shape in this example is demonstrated by the correlation between the loss of this shape and the appearance of pathologies, such as the Pelger–Huët anomaly [46,48]. Although beyond the scope of this review, it is important to note that dramatic nuclear changes also take place due to many other physiological reasons, such as in the nuclear fusion during mating, cell differentiation, and cell migration. In yeast, physiological changes in nuclear shape occur when haploid cells are in the presence of mating pheromones, which arrest cells in G1 in preparation for mating and thus for nuclear fusion, causing the nucleus to adopt a bilobed shape with a distinct pocket enclosing the rDNA; sphericity is also affected when the carbon source is changed or when cells are quiescent [49,50,51]. Furthermore, it has been described that during a mitotic delay, membrane synthesis is not hampered, giving rise to membrane expansions specifically at the NE adjacent to the nucleolus (Figure 1). These NE projections extend away from the bulk of the chromatin and are referred to as flares [52,53,54]. However, the most physiologically common and dramatic nuclear reshaping in the eukaryotic cell takes place during cell division. 

#### 1.2.1. Changes in the Nucleus during Mitosis

Organisms rely on the mitotic division to generate new cells, allowing growth and regeneration of tissues in multicellular organisms and population maintenance in single-cell organisms. Throughout evolution, there have been two main strategies for carrying out this function: open and closed mitosis. Open mitosis takes place in most eukaryotic cells, in which the NE is completely disassembled at the beginning of the M phase to allow microtubule attachment to chromosomes and their segregation. Once the chromosomes have reached each daughter cell, the NE reassembles around them to reconstitute the two daughter nuclei. On the other hand, closed mitosis is performed by certain species of fungi and protists in which the NE maintains its integrity so that chromosome segregation occurs entirely within the nucleus [55]. Noteworthy, there is a third strategy, known as semi-open (or semi-closed) mitosis, in which partial disassembly of the NE and/or NPCs allows both microtubule–chromosome interactions through nuclear pores and proper karyokinesis [56,57,58]. What all these strategies have in common is that they involve remarkable structural changes of the NE. In open mitosis, the NE needs to be completely rebuilt around all of the chromosomes to avoid deleterious effects, such as the appearance of micronuclei. In closed mitosis, the NE must increase its surface area and remodel to accommodate two nuclei, in addition to being severed and sealed (karyokinesis) at the end of the M phase [19,59,60,61]. In budding yeast, during exponential growth, the nucleus is mostly spherical until it gets stretched through the bud neck in mitosis for chromosome segregation (Figure 1). As this occurs, the nucleus acquires the shapes of a bowtie, an hourglass, and a dumbbell as the cell progresses from early to late anaphase. 

#### 1.2.2. Traits of Nuclear Shapes during Mitotic Arrests

Once chromosomes are duplicated and sister chromatids are properly aligned, it is time for the cell to equally divide sisters between the two daughter cells. The anaphase-promoting complex (APC) is a ubiquitin ligase composed of more than a dozen different subunits that exerts a variety of essential functions from yeast to humans that are regulated by specific APC cofactors [62]. When the APC binds to its regulatory cofactor Cdc20, the APC^Cdc20^ triggers a cascade of biochemical events that initiates anaphase. The details of these events are beyond the scope of this review, but it is important to note that blocking the APC function causes a metaphase arrest from yeast to mammalian cells [63]. To prevent the APC^Cdc20^ from segregating sister chromatids prematurely, there is a control mechanism called the spindle assembly checkpoint (SAC) that senses proper chromosome-microtubule attachment and tension [64].

There are several ways to achieve a mitotic arrest in metaphase. These arrests have traditionally been used to allow cell synchronization for easing interpretation of cell cycle studies. The three most commonly used arrest strategies are microtubule depolymerization by nocodazole (Nz), which triggers the SAC and inhibits the APC^Cdc20^, APC inactivation through conditional alleles (e.g., *cdc16-ts*), and Cdc20 depletion. The morphological traits between them are slightly different, presumably due to the lack of microtubules in Nz, but these methods arrest yeast as mononucleated dumbbell-shaped cells. In Nz-treated cells, the nucleus remains in the mother cell and presents a handbag structure, with the bulk of the chromatin in a large pocket and the rDNA locus as a handle (Figure 1) [65,66]. On the other hand, in APC inactive cells (either *cdc20* or *cdc16* defective), the nucleus tends to traverse through the bud neck, leaving the rDNA/nucleolus in the mother [66,67,68]. 

## 2. The Nucleolus and the rDNA

### 2.1. Features of the Nucleolus and the rDNA in the Budding Yeast

The nucleolus is the most prominent functional compartment within the nucleus. Transcription and processing of ribosomal RNAs (rRNAs), as well as ribosome assembly, take place in the nucleolus. In budding yeast, there is a single nucleolus characterized by a crescent shape at the periphery of the nucleus, occupying roughly one third of its volume, while in mammals, several nucleoli are located near the center of the organelle [41]. 

The nucleoli are assembled around rDNA loci, which are composed of the genes encoding the different rRNAs. The rDNA is organized as an array of tandem repeats, which can be physically clustered on a single chromosome or distributed over specific parts of the genome. In human cells, multiple rDNA arrays are shared around the five acrocentric chromosomes (13, 14, 15, 21, and 22) [69]. In contrast, the nucleolus of the budding yeast is assembled around a single rDNA locus on the right arm of chromosome XII (cXIIr) (Figure 2). The locus contains 150–200 copies of a 9.1 kb core repeat. Each repeat is in turn composed of a 35S transcription unit, two intergenic spacers (IGS1 and IGS2), and a 5S transcription unit lying in between [70,71]. The 35S transcription unit is highly transcribed by RNA polymerase I (Pol I), while the 5S is transcribed by the RNA polymerase III [72]. On average, half of the 35S transcription units is transcriptionally active, while the other half is silent. The Pol I, with a speed of 60 nt/s, accounts for up to 60% of transcriptional resources of the cell, while 50% of the total RNA polymerase II activity is devoted to genes encoding ribosomal proteins (RPs). Moreover, ribosome biosynthesis accounts for 50% of the total protein mass of the cell. Thus, ribosome biosynthesis is a highly demanding process for the cell, highlighting the central role of the nucleolus in cell growth and metabolism. Structural chromosome complexes such as cohesin and condensin, as well as silencing factors such as Sir2, are involved in rDNA compaction during the cell cycle, presumably helping to pack this piece of roughly 2 Mb of DNA into the crescent shape of the nucleolus at the nuclear periphery [66,73,74]. The rDNA locus is the last to segregate during the mitotic division, making cXIIr disjunction an indicator of the end of anaphase [65]. In addition to the general need for chromosome compaction for a safe mitotic conclusion, additional condensation of the rDNA by condensin is specifically required in late anaphase to allow cXIIr segregation [75,76,77,78]. 

Because the rDNA is repetitive and highly transcribed, it faces several challenges that put its integrity at risk. These include collision between replication and transcription machinery, with a greater risk for DNA double strand breaks (DSBs), as well as uneven sister chromatid exchanges during homology-directed repair (HDR) of DSBs. To counteract this, a replication fork barrier (RFB) blocks replication in the direction opposite to 35S transcription [79]. In addition, HDR is exquisitely regulated at this locus [70], favoring copy number maintenance of the repeats and avoiding intramolecular recombination, with the corresponding formation of extrachromosomal circles, which in turn have been linked to premature replicative senescence [80]. In addition, another way of enhancing the stability of the locus is by keeping the frequency of HDR lower than what would be expected. This is achieved through nucleolar exclusion of HDR proteins (but not sensors of DSBs) [81,82]. To support this, a subset of rDNA repeats is tethered to the NE by the CLIP and cohibin complexes, keeping the nucleolus tightly associated with the nuclear periphery. The CLIP complex is composed of the aforementioned Man1 homolog Heh1 and Nur1, both proteins inserted into the NE. Cohibin, on the other hand, connects the rDNA and silent chromatin to the NE by linking the rDNA-bound RENT complex with CLIP [31,36].

### 2.2. Shapes of the Nucleolus/rDNA during the Yeast Cell Cycle and NE Expansion

*S. cerevisiae* provides a unique genetic model to dissect chromosome behavior. Because the rDNA is the only chromosome structure visible under the microscope, the locus has been widely used to reveal chromosome dynamics, including condensation and segregation events [76,77,83,84,85].

The yeast rDNA undergoes drastic morphological modifications through the cell cycle, defined by changes in compaction and shape (Figure 3). During interphase (G1 and S), the rDNA acquires a loosely organized structure that has been referred to as puff, which evolves into a highly organized state on G2/M, especially during insults that delay or arrest cells at this cell stage. As described above, Nz exposure and consequent microtubule depletion entails a mid-M (metaphase) arrest where the rDNA adopts a remarkably complex loop shape. Because this loop is mediated by condensin, the loop has been considered a condensed state of the repetitive locus [73,74,86,87]. Additionally, cohesin, the Polo-like kinase Cdc5, and TORC1 (target of rapamycin complex 1) are also involved in the establishment and maintenance of the rDNA loop [87,88,89,90]. The relationship of TORC1, the master growth switch in response to nutrient availability, to nucleolar regulation is not surprising since transcription of the rDNA genes requires high levels of energy and cellular resources; thus, rDNA transcription must be shut down upon nutrient deprivation [91,92]. In addition, TORC1 is also apparently involved in rDNA chromatin structure by regulating the localization of RNA pol I and the Rpd3 histone deacetylase to rDNA [93]. 

Furthermore, a link between the nucleolus and NE morphological alterations has been established in recent years as abnormal NE structures induced by membrane overextension (nuclear flares) arise adjacent to the nucleolus, which may act as a membrane sink due to its ability to resist deformation [6,52,87,94,95]. This non-isometric NE expansion requires rDNA tethering to the NE during the mitotic arrests, as cells exhibit symmetric NE growth when tethering is impaired (CLIP/cohibin mutants). In addition, flare formation can be suppressed by reducing nucleolar size using low rDNA copy number strains [6]. Interestingly, these phenotypes differ from the flares observed in *cdc5* mutants, as these are neither affected by rDNA copy number nor rDNA tethering to the INM [53]. To further complicate matters, the link between NE expansion and the nucleolus may not be universal, not even within fungi. For instance, nuclear flares in the fission yeast *Schizosaccharomyces pombe* and its close relative *S. japonicus* are not associated with the nucleolus, suggesting that either nuclear domains for expansion have species-specific differences or they differ in the sites of new membrane synthesis [1,96]. Strikingly, besides the nucleolar ability to protrude upon NE excess, it has recently been reported that upon lipid stress, NE inward ingressions arise in association with the nucleolar subdomain, evidencing the plastic nature of the nucleolus in *S. cerevisiae* [97].

Not only the nuclear shape but also the nuclear size must be controlled, given that nuclear volume is maintained at 8–10% of cell volume in all organisms [98,99]. Previous evidence supports that nuclear and cytoplasmic volumes are correlated and maintain a relationship known as the karyoplasmic ratio (N/C). This ratio remains constant in yeast throughout the cell cycle as the cell increases in size, suggesting a mechanism that links nuclear and cellular volumes [98,100,101]. In experiments using *sec* mutants to arrest cell growth without disturbing the cell cycle, NE growth was not hampered and it was observed that the N/C ratio was kept constant by disturbing nuclear morphology in the form of flares (Figure 1) [102,103], suggesting that flares help to keep this ratio constant when NE proliferation is overstimulated [104]. Interestingly, in these scenarios, the nucleolus entirely occupies the most prominent flare. Nonetheless, and contrary to a general link between flares and constant N/C ratios, disruption of nuclear transport in *S. pombe* resulted in both an increased nuclear volume and shape defects due to NE excess [105].

## 3. Lipid Homeostasis and Its Relationship to Nuclear Shape

As the NE is mainly composed of phospholipids, lipid regulation is expected to be involved in nuclear size and shape. Indeed, NE expansion is dependent on phospholipid biosynthesis, as defects in these pathways result in misshapen nuclei in *S. pombe* and *S. japonicus* [96]. Likewise, deletion mutants of genes involved in the regulation of lipid homeostasis in *S. cerevisiae* also exhibit aberrant nuclear shapes [106,107,108]. Because the ONM is a continuum with the ER, one may think that ER over-expansion could affect nuclear morphology. However, even though there are mutants in which membrane overgrowth occurs in both compartments (*pah1*, *spo7*, *nem1*, and certain *sec* mutants), there are also examples in which they are independent, suggesting the existence of a barrier to membrane expansion between the peripheral ER and the NE [109,110,111].

Lipid precursors are used for phospholipid synthesis under nutrient-rich conditions to maintain membrane expansion during rapid proliferation, but are diverted to triacylglycerol (TAG) during starvation, which is packed in lipid droplets (LDs) for energy storage. Phosphatidic acid (PA) is the precursor for the synthesis of membrane phospholipids and LDs, and its fate is largely controlled by nutrient availability and the corresponding regulatory pathways [96,112]. The evolutionarily conserved lipin, Pah1 in budding yeast, generates diacylglycerol (DAG) from PA and has been proposed as a metabolic switch in response to growth and environmental signals, controlling the balance between lipid storage as LDs and membrane biogenesis [113,114,115] (Figure 4). Lipins are maintained inhibited in the cytosol by multisite phosphorylation, mediated by proliferative kinases such as Pho85, Cdc28, and PKA in yeast [116,117,118], and mTOR (mammalian TOR) and mitotic kinases in mammals [119,120,121]. Pah1 translocation onto the ER membrane is achieved after activation by dephosphorylation catalyzed by the Nem1-Spo7 transmembrane complex, which is subjected to TORC1 regulation [106,107,109,116,122,123]. Additionally, the activation of the DAG kinase Dgk1 counteracts the action of Pah1 by resynthesizing PA from DAGs, rerouting the pathway toward membrane biogenesis [108]. Cells lacking any of the proteins involved in the conversion of PA to DAG (Pah1, Spo7, and Nem1) exhibit aberrant nuclear shapes due to a reciprocal increase in membrane biogenesis, which gives rise to nuclear flares in the nucleolar region [95,107,112]. Furthermore, not only is NE morphology altered; cells with an impaired Nem1/Spo7-Pah1 axis exhibit defects in lipid droplet formation, vacuole morphology, and autophagy [124,125]. However, some of the phenotypes observed in the *pah1*Δ mutant are suppressed by the loss of Dgk1, suggesting that a proper balance of PA/DAG is required for lipid metabolism and cell fitness [108,126].

Work from the Cohen-Fix lab suggests that nuclear expansion in closed mitosis is independent of spindle elongation and occurs in response to a cell cycle cue signaling mitotic entry [52]. The flare induced by mitotic arrests shares similarities with nuclear expansions caused by misregulation of the yeast lipin/Pah1 pathway (including its regulators Nem1/Spo7) [95,126]. Using a mutant of this axis, Campbell et al. demonstrated that nucleolar expansion is not sufficient to drive flare formation. Additionally, *spo7*Δ cells combined with mutations in vesicle trafficking genes give rise to multiple flares around the entire nucleus, and not only at the nucleolus, indicating that the spatial confinement of a single nuclear flare is dependent on vesicle trafficking [104]. Thus, it has been proposed that there is a mechanism that favors nuclear elongation adjacent to the nucleolus to prevent disruption of nuclear inner chromosomal architecture, thus protecting the rest of the genome from NE over-expansion. Altogether, these works support the idea that the budding yeast NE has domains of different properties [128]. 

Lipid metabolic processes take place mainly at the ER surface, but recent studies in budding yeast have also spotted LD biogenesis at the INM adjacent to the nucleolus, providing a possible link between NE remodeling and lipid storage [94,129]. Barbosa et al. show a translocation event of the phospholipid:DAG acyltransferase Lro1 from the ER to the INM, resulting in a local TAG increase. Lro1 localization at the INM depends on cell cycle and nutrient availability signals, as it is excluded from the nucleolar territory during nuclear expansion, but recruited there upon starvation, promoting LD biogenesis [94,130]. Likewise, recent evidence also confirms nucleus–vacuole junctions (NVJs) as a site for LD budding in response to nutrient stress, consistent with previous evidence showing Pah1 near NVJs, and implicating the NE-vacuole tether Mdm1 in their biogenesis by interacting with enzymes of the fatty acid synthesis pathway [113,131].

## 4. The Vacuole, Nucleophagy, and TORC1

### 4.1. The Vacuole

Interactions of the NE with other organelles, including the vacuole and the ER, may affect the NE capacity for expansion [132,133]. Furthermore, the process of membrane remodeling frequently involves membrane contact sites, which are characterized by physical proximity between different endomembrane systems, as occurs at the interface between the nucleus and the vacuole [134].

The vacuole is the yeast equivalent of the lysosome, and is frequently associated with the nuclear envelope adjacent to the nucleolus [4,135]. Wild-type cells under normal growth conditions contain between one and four vacuoles, depending on the strain background [136]. The vacuole is critical for yeast viability, as it is required to perform multiple functions in response to cellular stressors and is also required for cell cycle progression; when vacuole inheritance to the daughter cell is prevented, the progeny is inviable [137]. Among its main functions, we encounter the turnover of cellular components via autophagy, storage of certain components, including phosphate, and the buffering of cellular pH, which plays a role in water and ion homeostasis [138]. Under nutrient deprivation, cells form a single vacuole, expanding their volume and probably facilitating autophagy [139]. The fusion of vacuoles require DAG and is hampered in lipin axis mutants because DAG synthesis is abolished, leading to vacuole fragmentation [140,141]. Strikingly, this phenotype is superseded by rapamycin-induced TORC1 inactivation, after which these mutants display a single vacuole, suggesting that autophagy itself is dispensable for vacuole fusion after TORC1 inactivation [124].

### 4.2. Nucleophagy

Autophagy is the general term used to define a group of processes that import a wide range of cargo, from portions of cytoplasm to entire organelles, into the vacuole/lysosome lumen for degradation by hydrolases. We mainly distinguish between macroautophagy, in which the cargo is packaged into a vesicle called an autophagosome, which then fuses with the vacuole, and microautophagy, in which the cargo is directly engulfed by the vacuole. These two processes can be subdivided into selective and non-selective macro- and microautophagy, respectively, depending on whether a specific cargo or bulk cytosol is affected. Specificity in macroautophagy is achieved through the binding of specific receptors to the specific cargo to be degraded and to the protein Atg8, which is involved in autophagosome formation. Autophagy initiation requires the Atg1 kinase and Atg13 to trigger phagophore formation followed by vesicle nucleation and recruitment of Atg8, among many other proteins (Figure 5) [142]. 

In organisms undergoing closed mitosis, because the NE remains intact throughout the cell cycle, misfolded and aggregated proteins must somehow be eliminated from the nucleus. Thus, cells must allow access to potential nuclear cargo by the autophagic machinery (Figure 5) [143,144,145]. Regarding the selective degradation of nuclear components, until recently, only microautophagic processes had been described in yeast. However, we can now differentiate between macronucleophagy and up to two microautophagic processes, referred to as piecemeal microautophagy of the nucleus (PMN) and late nucleophagy, respectively [144,146]. Macronucleophagy promotes the selective engulfment of nuclear cargos that comprise portions of NE, including proteins of the INM and ONM, as well as nucleolar proteins. However, it is not known whether nuclear partitioning of the cargo occurs prior to autophagosome formation or whether these two processes are synchronized [147]. Prolonged nitrogen starvation triggers macronucleophagy and, in turn, nuclear shape changes [142].

PMN was the first mechanism of nucleophagy described in budding yeast by Roberts et al. in 2003 and remains the best described. It takes place through direct contact between the two organelles at the NVJs, established by the Velcro-like interaction of the vacuolar protein Vac8 and the ONM protein Nvj1 [132,148]. Cargo is pinched from the nucleus directly into the vacuolar lumen by an electrochemical gradient generated by the vacuolar ATPase. [132,144,149,150]. In addition, there are other NE-vacuole contact sites established as sites for LD biogenesis that do not intervene in PMN; these contacts contain the tethers Mdm1, Nvj2, and Nvj3 [131,151]. One difference between the two autophagy modes is the size of the cargo, which is significantly larger in PMN [152,153]. Within the PMN process, two steps are distinguished: the initial protrusion of nuclear membrane at the NVJ and the late step of vesicle scission [149]. Through PMN, several cargos are degraded, including NE components and parts of the nucleolus containing pre-ribosomes and RNA, but importantly, NPCs and SPBs are excluded [154]. Incidentally, there is a specific autophagy pathway to selectively degrade NPCs upon TORC1 inactivation, which is divided into two branches referred to as NPC-phagy and nucleoporinophagy [133,155]. Because parts of the nucleolus are targeted for PMN, it is critical to prevent rDNA degradation. To avoid rDNA inclusion in the PMN nuclear projections, the CLIP and cohibin complexes drive the rDNA away from the NVJ upon TORC1 inactivation to physically separate it from nucleolar protein degradation [156,157]. This process of separation and repositioning of nucleolar proteins and rDNA is facilitated by rDNA condensation upon TORC1 inactivation [157]. The third type of nucleophagy described so far (and only recognized by a subset of authors in nucleophagy reviews) is referred to as late nucleophagy, which is a form of micronucleophagy that differs from PMN in time and space. First, and in contrast to PMN, it is independent of NVJs and some Atg proteins; and second, it is observed only after prolonged nitrogen starvation (24 h), whereas PMN is detectable in exponentially growing cells at a baseline level and at higher rates after much shorter nutrient deprivation (3 h). Interestingly, inhibition of late nucleophagy under nitrogen starvation conditions led to aberrant nuclear morphologies, yet they recovered a rounded shape upon re-exposure to nutrients [146]. 

### 4.3. TORC1

In contrast to other eukaryotes, in which several cues can activate autophagy, in yeast, nutrient deprivation is the main stimulus that triggers autophagy, although PMN is also active at basal levels during exponential growth under rich conditions [144,158,159,160]. The evolutionarily conserved TOR complex is the one that regulates growth by coupling nutrient sensing and cell metabolism. In yeast, unlike almost all other eukaryotes, there are two *TOR* genes, which in turn are present in two distinct complexes: TORC1, which contains either Tor1 or Tor2, and TORC2, which contains Tor2 exclusively. TORC1 is the one that responds to nutrients and rapamycin, and is the master regulator that controls protein and mRNA synthesis and degradation, ribosome biogenesis, nutrient transport, and autophagy. By contrast, TORC2 controls polarization of the actin cytoskeleton, endocytosis, and sphingolipid synthesis. TORC1 is considered to mediate the temporal control of cell growth [161].

TORC1 localizes mainly on the membrane of the vacuoles (of note, the yeast vacuole acts as a nutrient reservoir) and acts as a metabolic switch in response to nutrient availability. Accordingly, TORC1 is the primary sensor of nitrogen and amino acids, negatively regulating autophagy when they are present in sufficient concentrations, hence promoting cell growth (Figure 6). It does this by keeping Atg13 phosphorylated, which inhibits the downstream cascade triggered by this protein to assemble the Atg1 kinase complex [162]. In addition, it has been proposed that autophagy regulation by TORC1 requires at least a partial relocalization, in which it dissociates from the vacuolar membrane upon inactivation to allow Atg13 dephosphorylation [163]. 

Apart from inducing general autophagy, TORC1 inactivation promotes macro- and micronucleophagy through the Nem1/Spo7-Pah1 axis, as this axis is required to properly localize Nvj1 and the specific nucleophagy receptor Atg39 [124,156,164,165,166]. Additionally, the yeast lipin/Pah1 is recruited to the nuclear membrane domain in contact with the vacuole when glucose is consumed from the media in the post-diauxic shift, suggesting that it might play a role in NE remodeling to promote PMN [113]. Furthermore, other proteins involved in lipid biosynthesis (Tsc13 and Osh1) are targeted to the NVJs by Nvj1 and have been implicated in PMN, suggesting a link between lipid composition and the ability to form NE extensions toward the vacuole [148,149,150,167]. Additionally, TORC1 inhibition reduces nucleolar size in both interphase and mitosis, while also impinges on its morphology [87,93,168]. Cerulenin-induced inhibition of fatty acid synthesis prevents the formation of these NE extensions and impairs PMN, implying that lipid synthesis is needed to form these membrane protrusions [150,169]. In mammalian cells, mTOR phosphorylates lipin to keep it inactive and prevent it from binding to the nuclear membrane under favorable conditions [119,121]. However, yeast TORC1 helps to keep Pah1 inactive by preventing its dephosphorylation by Nem1; this is achieved by keeping the latter phosphorylated [123]. It has been suggested that abnormal nuclear morphology in *pah1* and *nem1* mutants may be exacerbated by impaired nucleophagy, as the cell could not recycle the excess of membrane created [170,171]. TAGs present in LDs act as donors for autophagosome membrane formation, suggesting that impaired TAG synthesis has a negative impact in autophagy induction. This has been observed in yeast mutants for TAG synthesis or lipolysis, which show inhibited autophagosome formation and macroautophagy after nitrogen starvation [172]. Furthermore, alteration of the lipin axis and impairment of DAG production compromises bulk autophagy, and especially nucleophagy, upon TORC1 inactivation [124,164].

## 5. Establishing a Morphological Axis Whereby the Vacuole Shapes the Nucleus, Nucleolus, and rDNA during Mitotic Arrests

Our group and others have described in *S. cerevisiae* that in a metaphase (mid-M) block triggered by Nz-driven microtubule depletion, the nucleus and the nucleolus drastically change their shape [52,173]. The spherical morphology is lost in favor of the appearance of a single finger-like projection that can bend to form a handle or loop. Furthermore, this projection affects the whole nucleolar domain, including the rDNA, ribosomal RNAs (rRNAs), and the rRNA-processing machinery. In our work, we further observed that the rDNA loop in a mid-M block makes the nucleus resemble a toroid, with the space under the rDNA loop occupied by the vacuole [173]. In addition, the establishment of this configuration depends on factors such as phospholipid biosynthesis, an active TORC1, functional vacuoles, and the absence of microtubules. However, the nuclear and nucleolar shapes are still severely affected in the presence of microtubules, with long nucleolar protrusions crossing the bud neck. We presented stepwise models of how the nucleus and the rDNA reshape during mid-M arrests based on both time course experiments and video microscopy, showing the formation of bilobed nuclei and how the position of the rDNA relative to the axis of NE expansion determines the final outcome, highlighting how seemingly similar rDNA handles and toroidal nuclei may actually have a rather different distribution of the NE (Figure 7).

Regardless of the complexity and variability of the nuclear shapes in the mid-M arrest, vacuoles appear as common remodelers [173]. Vacuoles are found associated from the early formation of the nucleolar protrusion to the very end products. Even in the long protrusions that cross the bud neck, vacuoles seem to pull the protrusion tip. Noticeably, other previous work had already reported small nuclear protrusions at NVJs [4]. In a remarkable percentage of bilobed and toroidal nuclei, the extent of the NE-vacuole relationship is such that the NE forms a ladle to accommodate vacuoles in it. The question that immediately comes to mind is the reason underlying this intimate association. We did not answer to this question in our work, and so remains open for future studies. However, we can speculate here. The fact that NE expansion depends on both an active TORC1 and lipid biosynthesis suggest that during mitotic blocks, the SAC, which ensures that cell cycle progression and chromosome segregation are efficiently stalled, does not target lipid synthesis and thus it is unable to pause the NE expansion required for nuclear extension in anaphase. Without the extension exerted by the anaphase spindle, the excess of NE reshapes the nucleus into morphologies that deviate from the perfect sphere, namely bilobes, toroids, and projections that grow outward. Perhaps, these abnormal shapes help in maintaining the volume ratio between the nucleus and the cytoplasm, as stated in previous sections. In this regard, vacuoles are connected to the NE through NVJs. An extensive use of these junctions may determine that NE overgrowth tends to engulf large vacuoles, whose shape is likely to be stiffer than the nucleus. Alternatively, vacuoles can regulate NE expansion. Perhaps there is an equilibrium between membrane synthesis for the NE expansion and nucleophagy for NE shrinkage, and this equilibrium is displaced toward the former as a consequence of a protracted TORC1 activity. Likewise, the location of TORC1 in the vacuolar membrane may efficiently couple its activity with NE expansion.

## 6. Conclusions and Future Perspectives

To sum up, the arrest of yeast cells in metaphase (mid-M) appears as a new condition that drastically remodels the nucleus, which acquires lobed and toroidal shapes. This remodeling particularly affects the nucleolus and, at the chromosomal level, the rDNA locus. Furthermore, the vacuoles appear as remodelers in these changes, which further depend on keeping TORC1 active and the biosynthesis of phospholipids unabated [173]. These findings, in turn, raise new questions and put into perspective other previous findings, especially those related to the chromosomal structure of the rDNA and nucleophagy. To start with the latter, and to tie in with the hypothesis we have just introduced in the previous section, it is intriguing that the vacuoles are found so tightly associated with the NE when TORC1 is active. One possibility is that the cells are taking steps to nucleophagy as the NE overgrows, but nucleophagy is nevertheless blocked by TORC1. Indeed, morphologically, the NE-vacuole wrapping may appear as over-extension of basal NVJs, which could later be corrected by PMN. In this sense, PMN might have an overlooked a role in the homeostasis of the NE in cycling cells, not only during starving conditions, which is abnormally blocked during mitotic arrests. This hypothesis should be further investigated in the future, since it is evident that impairing nucleophagic mechanisms must have implications for the organism’s health, as has been reported in many human pathologies, including cancer, type II diabetes, several central nervous system diseases, such as Parkinson’s disease, and in the aging process as a whole [160,174,175,176]. In addition, it would be interesting to determine the presence and distribution of NE proteins (NPCs, LINC and SPBs) in these protrusions and whether they are meant to be targeted for nucleophagy later on, in case the previous hypothesis is confirmed. Noteworthy, NPCs are barely present in the NE ladle that engulfs vacuoles [173].

Another important implication of non-round nuclear shapes is how chromosomes are spatially reorganized and how this affects gene regulation. Although we did not determine the spatial organization for chromosomes other than XII, it is expected that the new extreme shapes influence this extensively. Hence, nuclear shape should be included in studies that attempt to describe the topology of chromosomes. In this context, it is worth mentioning that in a previous work where chromosome positions were established by chromosome conformation capture (Hi-C) in Nz-arrested cells, the authors found that regions flaking the rDNA do not interact with each other [177]. They attempted to fit this to a model in which the nucleus was a sphere. However, this result is easier to interpret and fits well with the bilobed nuclei we have described (Figure 7) [173]. Similarly, the association of NE extensions with the nucleolus might provide a mechanism for the changes in the rDNA structure observed in mitotic arrests. The fact that this structure also depends on cohesin and condensin [74,87], but is independent of rDNA-NE tethering by the CLIP/cohibing [173], raises questions about the interplay between all these players. Finally, it is not lost on us that the new nuclear morphologies and the close interaction with the vacuoles can have profound consequences in the anaphase that follows the release from the mitotic arrest. 

One of the nucleus–vacuole associations we found, the toroidal nucleus with the vacuole in the center, has also been described in human cell lines as well as in primary tumor cultures [178]. Nevertheless, in human cell lines, this outstanding morphology stems from lysosome malfunction and is seen post-mitotic, i.e., once the NE is reestablished in telophase, which appears to differ from what we observed in yeast [173]. It would be interesting to compare the similarities and differences of these phenomena, since studies in yeast could simplify our experimental approach to this remarkable tumor-related phenotype. 

The induction of changes in the nuclear shape are meant to have important implications for cell physiology and genetic stability. Another critical question is whether nuclear reshaping can be protective rather than deleterious, at least for a subset of stressful conditions. In addition, the fact that bilobed and toroidal nuclei can be achieved in yeast with drugs that arrest mitosis opens new avenues for the development of better antifungal and antiparasitic therapies, with plausible combinations of drugs targeting the cell cycle, TORC1, and lipid metabolism against organisms that undergo closed mitoses. These can also be lines of research to pursue in the future.

Altogether, we can envision how intricate the homeostasis of the NE is, taking into account processes such as lipid biosynthesis and the degradation of nuclear components via nucleophagy, the connection to other organelles such as the vacuole, which in turn are regulated by different pathways such as TOR-mediated surveillance. This brings forward that the “healthy” round nucleus is achieved and maintained through complex networks that are far from the simple intuition based on mere physical properties of the sphere. 

## Figures and Tables

**Figure 1 ijms-24-09829-f001:**
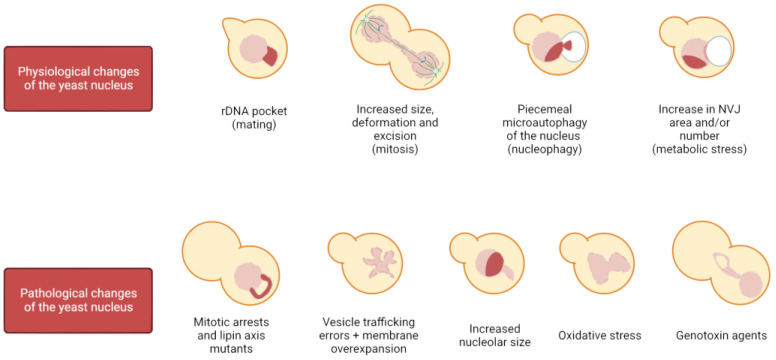
Events that alter the spherical shape of the nucleus. On the top, the yeast nucleus experiences several changes in morphology due to physiological reasons, among which we encounter the appearance of a distinct rDNA pocket when cells prepare for mating, the increase in size and elongation during mitosis, teardrop protrusions toward the vacuole in PMN, and an increase in NVJs size and/or number under stress. At the bottom, other events trigger non-physiological changes in nuclear morphology, such as the appearance of flares (NE elongation confined to the nucleolar area) in mitotic arrests and lipin axis mutants, non-localized deformations when vesicle trafficking is disrupted under membrane overexpansion conditions, upon oxidative or genotoxic stress, and when nucleolar volume is increased.

**Figure 2 ijms-24-09829-f002:**
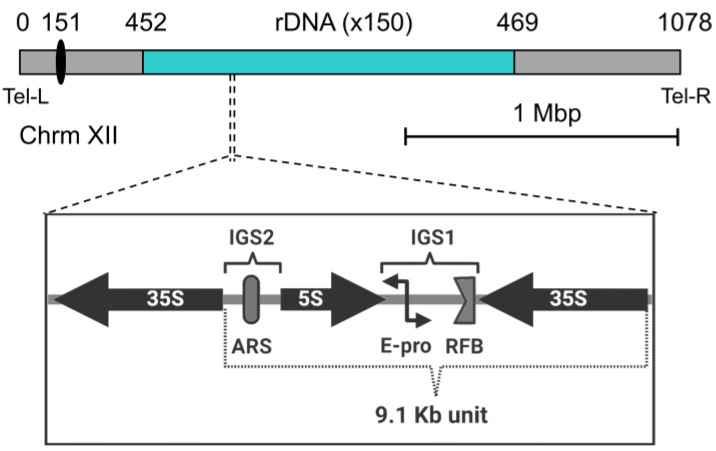
Genomic architecture of the budding yeast rDNA. The locus (in cyan) extends along most of the right arm of chromosome XII (to scale, considering a locus of 150 rDNA repeats). Numbers above indicate Saccharomyces Genome Database (SGD) coordinates (0, left arm telomere; 151, centromere [depicted as a black oval]; 452, proximal rDNA flank; 469, distal rDNA flank; 1078, right arm telomere). Below, the unscaled schematic represents the organization of the basic 9.1 kb rDNA repeat. 35S and 5S, genes for the corresponding pre-rRNAs; IGS 1 and 2, intergenic spacer 1 and 2; ARS, autonomous replication sequence; RFB, replication fork barrier; E-pro, a cryptic non-coding bidirectional promoter.

**Figure 3 ijms-24-09829-f003:**
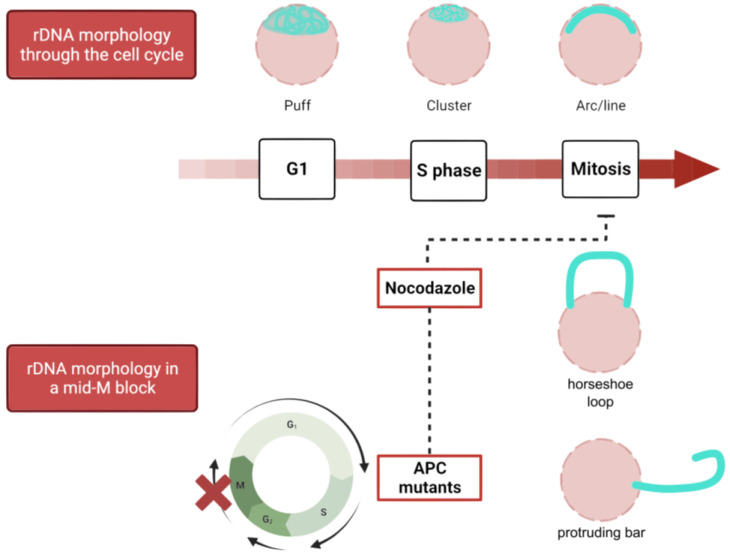
Morphological changes of the rDNA through the cell cycle and in mid-M blocks. The rDNA (in cyan; the rest of the nucleus is depicted in pink) acquires different morphologies during a normal cell cycle, from a disorganized puff in interphase which gradually transforms into a condensed and organized arc or line in mitosis. Nevertheless, in mid-M blocks caused by the microtubule depolymerizing drug nocodazole or conditional APC mutants, the rDNA develops nucleolar extensions around the rDNA array, causing the locus to protrude from the main nuclear mass as a horseshoe, a bar, etc. APC, anaphase-promoting complex.

**Figure 4 ijms-24-09829-f004:**
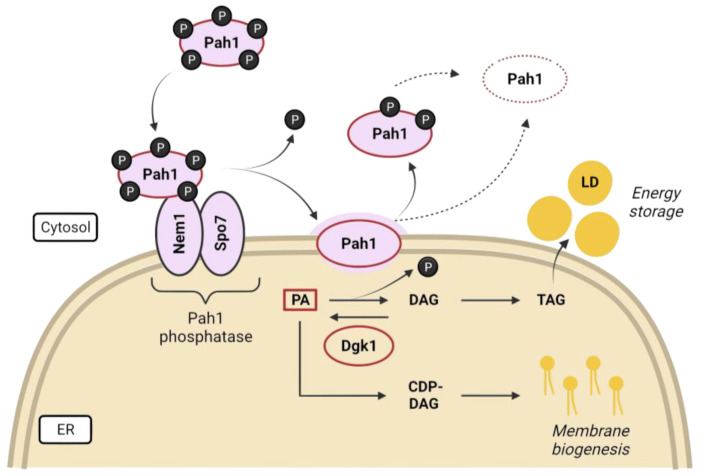
Regulatory network of the yeast lipin/Pah1. Phosphatidic acid (PA) is the central precursor in lipid biosynthesis. Pah1, the PA phosphatase, remains phosphorylated (inactive) in the cytosol under favorable growth conditions so that phospholipid biosynthesis for membrane expansion is promoted via the CDP-DAG pathway. However, when nutrients are scarce, the Nem1-Spo7 complex embedded in the ER membrane dephosphorylates Pah1 to activate it and catalyze the conversion of PA to DAG, which is further acylated and converted to TAG and stored in lipid droplets (LD). Dephosphorylated (or partially dephosphorylated) Pah1 is targeted for proteasomal degradation (adapted with permission from [127]).

**Figure 5 ijms-24-09829-f005:**
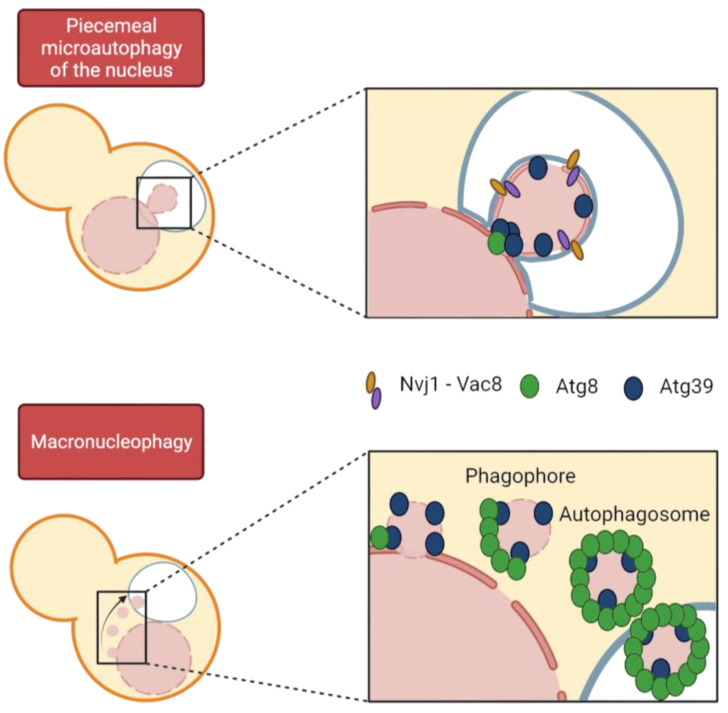
Modes of nucleophagy in budding yeast. Both macro- and micronucleophagy depend on the interaction of Atg8 and the specific cargo receptor Atg39, yet Nvj1 is only detected in PMN vesicles. Macronucleophagy involves autophagosome formation, coated by Atg8, which recruits the rest of the autophagic machinery. By contrast, PMN vesicles are not coated with Atg8, except for a focus at the neck of nuclear protrusion into the vacuole. Remarkably, PMN cargo is larger than that of macronucleophagic vesicles.

**Figure 6 ijms-24-09829-f006:**
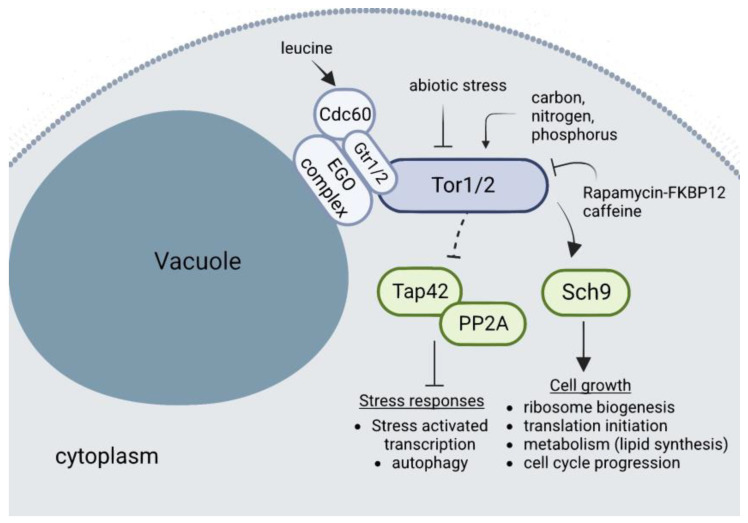
Details of TORC1 signaling in budding yeast. The 1.2 MDa TORC1 is composed of the Lst8, Kog1, Tco89, and one of the two Tor proteins (Tor1 or Tor2). The EGO complex anchors TORC1 to the vacuole membrane. When cellular metabolites are present, the EGO complex act as a signalling hub by transducing signals to TORC1. The Rag family GTPases Gtr1 and Gtr2 form the functional end of the EGO complex. The cytosolic leucyl tRNA synthetase (Cdc60) is activated by the presence of the amino acid leucine and activates in turn the EGO complex. TORC1 activation leads to the phosphorylation of the downstream target protein Sch9, the yeast ortholog of the mammalian S6 kinase, and inhibition of the Tap42/PP2A proteins. Arrowheaded lines indicate activation, while a straight headed line indicates inhibition. For clarity, the TORC1 and the EGO complexes and their bound Gtr subunits are simplified.

**Figure 7 ijms-24-09829-f007:**
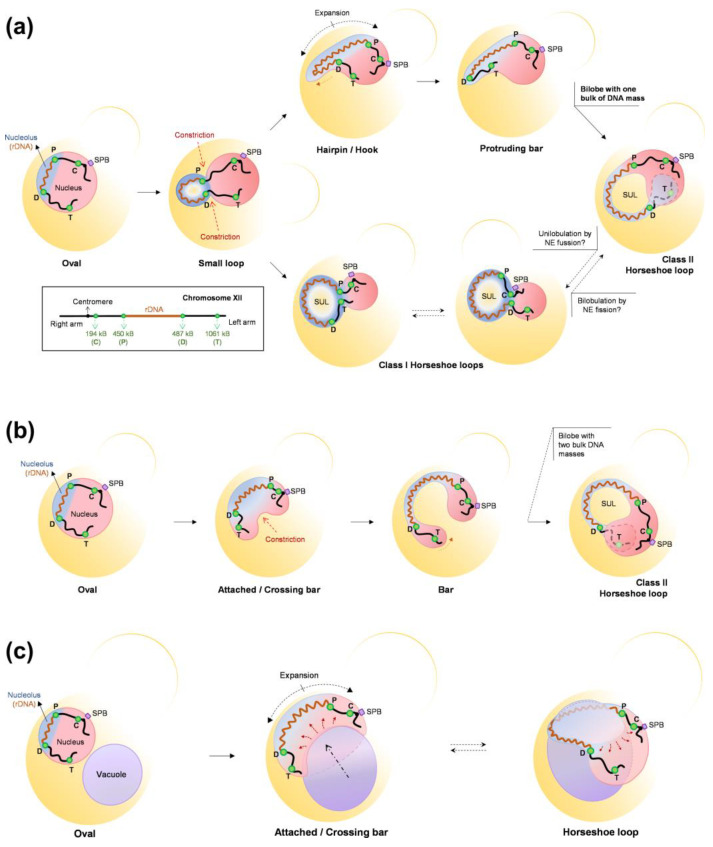
Paths of NE expansion, rDNA loop formation, and the role of the vacuole. (**a**,**b**) Paths of nucleolar envelope expansion and nuclear constrictions that shape rDNA and nuclear morphologies during the Nz-induced mid-M arrest. In (**a**), the NE grows outward to form the nucleolus bud after the ensuing basal constrictions. Two branches can then be drawn. In the upper branch, the nucleolus grows outward as a finger-like projection, which ends up pulling the distal cXIIr arm into the projection; the resulting protruding bar creates a bilobed nucleus that bends until both lobes overlap, giving rise to the class II horseshoe loop (dashed lines for cXIIr telomeric regions indicate that they are behind the rest of the nucleus). In the lower branch, the nucleolar NE continues expanding symmetrically in all directions, making possible the flourishing of the class I horseshoe loop, characterized by a unilobed nucleus and an NE ladle at the SUL. It remains to be determined whether classes I and II are interchangeable through NE fission and fusion events. In (**b**), the NE expands laterally, creating an opposing nuclear constriction so that the expanding rDNA bar/handle connects a newly formed bilobed nucleus; the bar then bends until both lobes touch each other, giving rise to another presentation of the class II horseshoe loop. (**c**) Schematics of how the vacuole may shape the SUL NE ladle. The vacuole can act as a scaffold for the NE to grow at the nucleolar region, perhaps from nucleus–vacuole junctions (NVJs). Vacuolar engulfment by the NE may explain the ladle shape of the SUL in horseshoe rDNA loops. The rDNA array is depicted as a dark red spring, the rest of the chromosome XII (cXII) as a black thick line, the nucleolus is in blue, and the main nuclear DNA mass is in light red. Four cXIIr reference regions are indicated by green dots (C: next to the centromere, P: proximal or centromeric rDNA flank, D: distal or telomeric rDNA flank, T: next to telomere; a cXII schematic is also depicted on the left bottom corner of panel (**a**)). Black arrows indicate morphological transitions, dashed black double arrows indicate the direction of NE expansion, and red dashed arrows the motion of nuclear subdomains during transitions. The budded yeast cell is outlined in yellow, and the mother cell is filled in degraded yellow. SUL: Space under the rDNA loop; SPB: Spindle Pole Body. Further details can be found in [173].

## Data Availability

Data sharing is not applicable to this article.

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
