# Peer review of "A Yeast Mitotic Tale for the Nucleus and the Vacuoles to Embrace"

_ijms, 2023, doi:10.3390/ijms24129829_

Round 1

Reviewer 1 Report

This rather diffuse review discusses many aspects of nuclear organization, with primary focus on Saccharomyces cerevisiae. The title refers to interesting findings from the authors’ lab on the distortions of yeast nuclei that occur in cells arrested in mitosis, but these are only discussed in the last section (section 7). Much of the rest of the review deals with topics that do not cohere into any single theme and are in most cases not needed in order to discuss the mitotic nuclear shape. Perhaps a much shorter perspective incorporating only the relevant material and having a much tighter focus would be of interest to readers.

Positive aspects: 

·      Where I have familiarity with the cited literature, the authors’ descriptions seem accurate.

·      Most of the Figures and Figure Legends were well made and clear (an unfortunate exception being Fig. 8, which seems overly complicated).

·      The section 6 on nucleophagy was interesting (although its relation to section 7 was not well explained).

Negative aspects:

·      Many sections (>50%) cover material not directly relevant to the main issue (e.g. rabl organization, LINC components, SMC complexes, detailed organization of rDNA repeats, specific regulatory pathways for membrane synthesis, and indeed most gene names listed in various parts). 

·      While a consideration of the relevance of yeast nuclear shape to findings in other systems is worthwhile, references to other systems were scattered throughout the review without obvious reason and were frequently confusing or distracting rather than enlightening.

·      The material that WAS most relevant to nuclear shape (e.g. distortions due to excess membrane, nuclear-vacuole junctions, nucleophagy) was not organized in a manner that made clear how it directly pertained to the mitotic distortions. The ideas are perhaps there, but diffuse and not well explained.

·      It appears to be an article of faith for the authors that nuclear shape is important and that we should all care about it. But they did not really explain why, and some of the arguments seem dubious. 

·      The writing was often ungrammatical, and overly florid for scientific English. The meaning was often difficult to understand.

In summary, while the focus here involves a potentially interesting phenomenon, the review in its current form is far too long and does not synthesize information in a useful way. To make this review of value to its readers would require very extensive shortening, focusing, and editing.

The writing was often ungrammatical, and overly florid for scientific English. The meaning was often difficult to understand.

Reviewer 2 Report

This review is generally good, and I enjoyed reading it. The authors provide a very detailed and thorough discussion of different aspects of yeast nuclei and the potential importance of dynamics of nuclear morphology. Change in nuclear shape is not only related to cell division but also correlates these changes to the nucleolus and vacuole. They discuss that vacuole shapes nucleolus and rDNA in mitotic arrested cells which depends on phospholipid synthesis and an active TORC1 (Target of rapamycin complex 1). Based on recent studies, the interplay between nuclear morphology and vacuoles (lysosomes) may provide a better understanding of cellular physiology and genetic stability of cells (e.g. cancer or diseased or stressed cells).

I have the following suggestion for the author’s consideration

A section on "conclusions and future directions" would be a valuable addition to the review. This section could summarize the main findings of the review and highlight the most important implications for future research. it would be appropriate to draw from the existing paragraphs of the review and reorganize them into this section.

Round 2

Reviewer 1 Report

This has been improved in many ways, though some of the original problems remain to a lesser degree.

This is improved, but still occasionally hard to follow.
